# Getz Ice Shelf melt enhanced by freshwater discharge from beneath the West Antarctic Ice Sheet

Wei Wei[1], Donald D. Blankenship[1], Jamin S. Greenbaum[1], Noel Gourmelen[2], Christine F. Dow[3], Thomas G. Richter[1], Chad A. Greene[4], Duncan A. Young[1], SangHoon Lee[5], Tae-Wan Kim[5], Won Sang Lee[5], and Karen M. Assmann[6]

[1]Institute for Geophysics and Department of Geological Sciences, Jackson School of Geosciences, University of Texas at Austin, Austin, TX, United States.
[2]School of Geosciences, University of Edinburgh, Edinburgh, United Kingdom.
[3]Department of Geography and Environmental Management, University of Waterloo, Waterloo, Ontario, Canada.
[4]Jet Propulsion Laboratory, California Institute of Technology, Pasadena, California, United States.
[5]Korea Polar Research Institute, Incheon, South Korea.
[6]Department of Earth Sciences, University of Gothenburg, Gothenburg, Sweden.

**Correspondence:** Wei Wei (wwei@utexas.edu)

**Abstract.** Antarctica's Getz Ice Shelf has been rapidly thinning in recent years, producing more meltwater than any other ice shelf in the world. The influx of freshwater is known to substantially influence ocean circulation and biological productivity, but relatively little is known about the factors controlling basal melt rate or how it is spatially distributed beneath the ice shelf. Also unknown is the relative importance of subglacial discharge from the grounded ice sheet in contributing to the export of 5 freshwater from the ice shelf cavity. Here we compare the observed spatial distribution of basal melt rate to a new sub-ice shelf bathymetry map inferred from airborne gravity surveys and to locations of subglacial discharge from the grounded ice sheet. We find that melt rates are high where bathymetric troughs provide a pathway for warm Circumpolar Deep Water to enter the ice shelf cavity, and that melting is enhanced where subglacial discharge freshwater flows across the grounding line. This is the first study to address the relative importance of meltwater production of the Getz Ice Shelf from both ocean and subglacial 10 sources.

## 1 Introduction

The Getz Ice Shelf (Getz, herein) in West Antarctica is over 500 km long and 30 to 100 km wide; it produces more freshwater than any other source in Antarctica (Rignot et al., 2013; Jacobs et al., 2013; Assmann et al., 2019), and in recent years its melt 15 rate has been accelerating (Paolo et al., 2015). The fresh, buoyant water that emanates from the Getz cavity drives regional and global ocean circulation (Nakayama et al., 2014; Jourdain et al., 2017; Silvano et al., 2018) while providing critical nutrients for biological production (Raiswell et al., 2006), but little is known about the origins or sensitivities of this major freshwater

source. Specifically, the variability of freshwater from ice shelf melt has not been modeled due to poorly constrained bathymetry beneath the ice shelf, which has resulted in a poor understanding of how water circulates throughout the ice shelf cavity. And to date, despite the major role that freshwater from Getz plays in the Southern Ocean, no studies have considered the contribution of subglacial meltwater that originates beneath grounded ice, nor its role in influencing the circulation and melt patterns beneath the ice shelf.

To understand the potential pathways for warm ocean water to enter in the Getz cavity we conducted a bathymetric survey using a ship-based, gravimeter-equipped helicopter. As part of a collaboration between the University of Texas Institute for Geophysics (UTIG) and Korea Polar Research Institute (KOPRI), an AS350 helicopter was outfitted with an aerogephysical instrument suite adapted from a design which has previously been operated from fixed-wing aircraft (Greenbaum et al., 2015; Tinto and Bell, 2011). Operating from the RVIB Araon off the Getz coast (see Supplementary Fig. S1), the survey covered areas between Dean Island and Siple Island located in the west of Getz (green lines in Fig. 1) and crossed existing coast parallel Operation IceBridge (OIB) lines (blue lines in Fig. 1). Gravimetry from helicopter platform can achieve higher resolution of 3 km than conventional fixed wing surveys with resolution of 4.9 km due to lower flying speed of helicopter. This is the first ever demonstration of the technical and logistical feasibility of gravity observations from a helicopter operating from a ship at sea to obtain high resolution gravity data over an Antarctic ice shelf.

We used the airborne gravity data to infer the bathymetry beneath Getz (see methods on the bathymetry inversion approach). We also developed a new high-resolution map of Getz basal melt rates using satellite radar altimetry data from the years 2010 to 2016 (see methods on the observed basal melt rate) to understand where ice shelf melt may be correlated with underlying bathymetry. By comparing locations and rates of melt with our new understanding of the Getz cavity bathymetry, we gain first insights into where ice shelf melt is dominated by contact with Circumpolar Deep Water (CDW) and where bathymetry blocks the flow of CDW. We also considered the potential role of subglacial discharge as a mechanism that can cause locally enhanced melt rates (Le Brocq et al., 2013; Marsh et al., 2016). We used the Glacier Drainage System (GlaDS) model, which simulates co-evolution of subglacial distributed and channelized drainage networks that have been demonstrated to correspond well with geophysical data of basal water systems (Dow et al., submitted). We applied this model to estimate the production rate and spatial distribution of subglacial meltwater (Werder et al., 2013; Dow et al., 2016, 2018) (see methods on the subglacial hydrological model). We then compared the spatial distribution of observed ice shelf melt to locations and flux rates from subglacial discharge locations predicted by GlaDS.

## 2  Methods

We present three types of data in the study: the spatial distribution of basal melt rate (see Fig. 2b and Fig. 3) , bathymetry inferred from airborne gravity surveys (shown in Fig. 2a), and locations of subglacial discharge (see Fig. 3) from the grounded ice sheet. We inferred the bathymetry of Getz using profile gravity inversions with the Geosoft GMSYS software. The subglacial hydrological analysis was generated by the two-dimensional GlaDS (Glacier Drainage System model) (Werder et al., 2013). The observed basal melt rates were computed using a mass conservation approach from Jenkins (1991) and Gourmelen

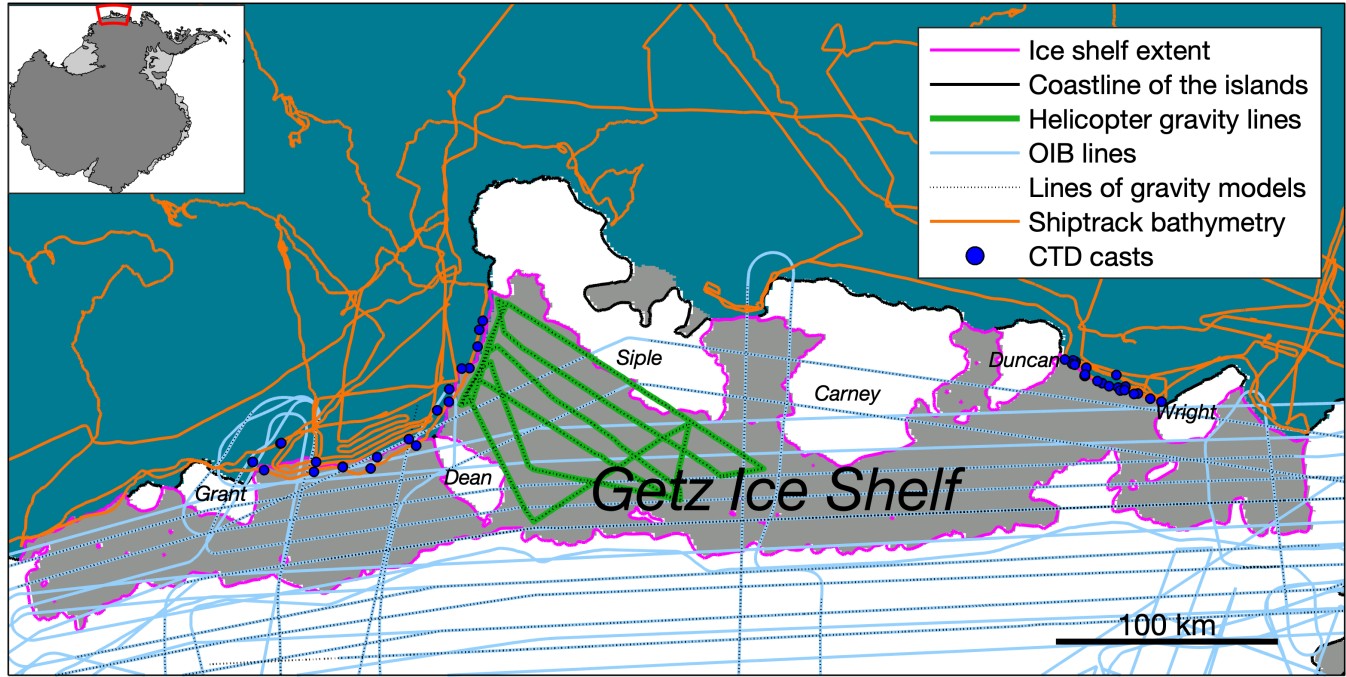

**Figure 1. The geographic location and data coverage of Getz.** The blue area is ocean. The gray area is Getz Ice Shelf. The white area is grounded ice. Coloured lines and marks denote the ice shelf extent (Mouginot et al., 2017), helicopter gravity data, and NASA OIB data (Cochran and Bell, 2010), shiptrack bathymetry (Nitsche et al., 2007) and CTD casts (Locarnini et al., 2013). This plot is generated from Antarctic Mapping Tool (Greene et al., 2017).

et al. (2017b), and corrected for melting driven by warm ocean waters using ice bottom elevation and nearby ocean temperature profiles (Holland et al., 2008).

## 2.1 Helicopter gravity data acquisition

The gravity data used in this paper was acquired aboard two aircraft types, one fixed-wing aircraft and one helicopter. Fig. 1
shows the data coverage. The OIB data (Cochran and Bell, 2010) was acquired using the Sander Geophysics Limited (SGL) AIRGrav system aboard NASA's DC-8. More details of this airborne geophysical platform can be found in the literature (Cochran and Bell, 2012; Cochran et al., 2014). The helicopter based data was acquired using a Canadian Micro Gravity GT-1A in a collaboration between the University of Texas Institute for Geophysics (UTIG) and Korea Polar Research Institute (KOPRI). Fig. S1 shows the helicopter gravity data acquisition platform on the icebreaker Araon. Three dedicated aerogeo-
physical flights were accomplished in one day of helicopter operations from the Araon while off the coast of the Western Getz, acquiring about 1200 line-kilometers of data. The gravity anomaly (Fig. S2) suggests the effectiveness of combining OIB and helicopter data. The observed gravity anomaly ranges from -60 mGal to 30 mGal (Fig. S2). The high anomaly strongly corre-

lates with the ice rises and grounded icebergs. Large positive gravity anomalies of up to 30 mGal are consistently found over Grant Island, Dean Island, Siple Island, and Wright Island. The areas between ice rises correspond to low gravity anomalies.

Both survey data sets show similar repeatability statistics with ∼1.4 to 1.6 mGal root mean square (RMS) in the differences at crossovers between lines both internally in each set and between sets. The ship based UTIG/KOPRI gravity set did not have an absolute gravity tie so that entire survey set was level shifted to minimize the difference in the mean of crossovers with the OIB data; no other adjustments were done.

## 2.2   Bathymetry inversion approach

The gravity data is inverted for depth of targets using the GM-SYS Profile Modeling, a 2D gravity modeling and inversion module in Geosoft. In the forward modeling mode, the module computes the gravity response from a polygon approximated irregular target model (Talwani et al., 1959). In the inversion mode, the polygon approximated model is adjusted iteratively to best fit the observed gravity data. Getz is pinned on an array of islands and peninsulas, so our bathymetry inversion is well constrained by the location of the ice rises and the peninsulas. The bathymetry model is updated iteratively until the difference

between modeled gravity and observed gravity values is minimized (convergence limit = 0.1 mGal, 0.1 mGal is the standard error of observed gravity data). To better condition the inversion process, we fix the top and bottom of the ice layers, whose depth and topography are obtained from radar data. Similar approaches to infer bathymetry from airborne gravity data have been applied in many regions of Antarctica (Tinto and Bell, 2011; Cochran and Bell, 2012; Muto et al., 2016; Millan et al., 2017; Greenbaum et al., 2015).

We first use the gravity data from grounded ice lines to invert for bedrock densities. For those areas covered by the grounded ice lines, we assume a three-layer model: a solid ice layer with density of $917 \, \text{kg} \cdot \text{m}^{-3}$ of known thickness over a bedrock layer, whose density is our free parameter; the third layer is the upper mantle with a density of $3300 \, \text{kg} \cdot \text{m}$ at a depth of 20 km. The top, bottom, and thickness of the ice layer is obtained from OIB measurement and thus fixed throughout the inversion. We start the inversion with an initial guess of granitic rock density value $2.75 \, \text{kg} \cdot \text{m}$ since west of the survey area has granite outcrop

(Mukasa and Dalziel, 2000).

   The gravity data from floating ice lines is used to invert for the bathymetry under Getz. We use a four-layer model: the first layer is an ice layer with density of $0.917 \, \text{kg} \cdot \text{m}$ with a known depth; the second layer is a seawater layer with a density of 1030 $\text{kg} \cdot \text{m}$, the depth of this layer is our free parameter; the third layer is a bedrock layer with density inferred from grounded-ice-line gravity data; the fourth layer is the upper mantle with a density of $3300 \, \text{kg} \cdot \text{m}$ at a depth of 20 km. The lines are processed one

by one starting from those that are closer to grounded ice lines. The bathymetry model is updated iteratively until the difference between modeled gravity and observed gravity values is minimized (convergence limit = 0.1 mGal, 0.1mGal is the standard error of observed gravity data). The polygon densities applied in this region is in Fig. S3. The constructed 2D models can be found in Fig. S4.

   The different 2D bathymetric profiles are merged through the minimum curvature gridding method, provided by the Grid

and Images module from Geosoft Oasis montaj. The details of the minimum curvature method can be found in Briggs (1974). The mesh size of the interpolated grid is 1792 m. To prevent aliasing and high frequency signals, we increase the low-pass

de-sampling factor (i.e., the number of grid cells that are averaged). This factor (set to 3) removes high frequency signals since it acts as a low-pass filter by averaging all point into the nearest cell. The distance between grid cells and a valid point greater than the blanking distance (set to 2000 m) are blanked out in the final grid. However, gridding could still introduce artifacts at the intersection points. We calculate the offsets at the profile intersections, and the average offset is about 20 m. The final derived bathymetry (Fig. 2a) includes the Getz Ice Shelf bathymetry from gravity inversion and offshore bathymetry from IBCSO (International Bathymetric Chart of the Southern Ocean) (Arndt et al., 2013).

We follow the uncertainty estimation approach from Greenbaum et al. (2015). We compare the inversion with the geometry of the grounded ice as a measure of the uncertainty beneath the floating ice assuming that the bed roughness under grounded ice and floating ice are similar. Our estimated Root Mean Square Error (RMSE) between the ice bottom measured by radar and sampled from the bathymetry model is about 246 m and the mean offset between the two is about 44 m (see Supplementary Fig. S5). We also compare the overlapping points where the gravity lines intersect with the shiptrack (Nitsche et al., 2007). The Root Mean Square Error (RMSE) between the ship measured bathymetry and sampled from the bathymetry model is about 121 m, the mean offset between the two is about 32 m (see Supplementary Fig. S5).

We do not include significant geological or sedimentary signatures in our model since we have insufficient magnetic analysis over Getz. But our methods do account for local geological heterogeneity (see Supplementary Fig. S3). Published interpretations within ASE (Amundsen Sea Embayment) imply that we should not expect a significant crustal thickness gradient or sedimentary basin lies beneath the Getz Ice Shelf (Gohl et al., 2013). Therefore, we expect sediments near the grounding line being scoured away as seen in other ice shelves of ASE (Gohl et al., 2013; Cochran et al., 2014). However, if the sediment is present, it will cause the gravity-derived bathymetry to be deeper than the actual seafloor. If we are correct and no significant geological structure underlies the Getz then the existence of sediments will shift the bathymetry to be deeper but will not change the shape of the bathymetry, and thus will not affect our conclusions.

### 2.3 Subglacial hydrological model

The subglacial hydrological analysis is generated by the two-dimensional GlaDS (Glacier Drainage System model) (Werder et al., 2013). Distributed flow occurs through linked cavities that are represented as a continuous water sheet of variable thickness. Channels grow along finite element edges and exchange water with the adjacent distributed system, as part of a fully coupled 2D drainage network. The model is run to the steady state over 3000 days with primary outputs being channel discharge over the domain and the grounding line into the Getz cavity. Topography inputs are from airborne radar data; basal velocity is estimated as 90% of MEaSUREs surface velocity data (Rignot et al., 2017); basal conductivity is assumed constant following other applications of GlaDS in Antarctica (Dow et al., 2016, 2018). Water input rate is set as constant (both spatially and temporally) at 10 mm·yr$^{-1}$ following geothermal flux rate calculations (Pattyn, 2010).

### 2.4 Observed basal melt rate

The observed ice-shelf basal melt rates are computed using a mass conservation approach from surface elevation, surface mass balance, ice velocity and ice shelf thickness (Jenkins, 1991; Gourmelen et al., 2017b), using the relation (Jenkins, 1991;

 Gourmelen et al., 2017b)

$$-\left(1-\frac{\rho_{\text{ice}}}{\rho_{\text{ocean}}}\right)\dot{m} + \text{SMB} = \frac{\partial S}{\partial t} + S\nabla \cdot \mathbf{u}, \tag{1}$$

where $\rho_{\text{ice}}$ is ice density of 917 kg·m$^{-3}$, $\rho_{\text{ocean}}$ is the ocean density of 1028 kg·m$^{-3}$, $\dot{m}$ is basal melt rate, SMB is surface mass balance, $S$ is surface elevation and $\mathbf{u}$ is ice velocity. SMB is obtained from output of the regional atmospheric climate model RACMO2 (Van Wessem et al., 2016). We derive the rates of surface elevation change from a new elevation dataset, which is generated by the CryoSat-2 interferometric-swath radar altimetry from 2010 to 2016. Ice velocity is acquired from radar observation of the European Space Agency Sentinel-1a mission. A detailed discussion of the methodology can be found in Gourmelen et al. (2017a). The observed melt rate of Getz is shown in Fig. 2b.

## 2.5   Non-discharge melt rate

We modeled the melt rates over Getz that are expected to result from the in situ farfield ocean temperature. We refer to this modeled melt rate as non-discharge melt rate through this paper since it does not consider any potential impact from local subglacial discharge. Melt rate shown in Fig. 2b are dominated by ocean forcing. To the first order, melt rates are visibly related to ice basal depth, and accordingly we note that melt rates are high where the draft of the ice shelf dips below the ∼500 m depth of the thermocline. As our interest is in exploring the possible mechanisms of melt beyond the first-order effects of ocean forcing, Fig. 3 shows melt rate residuals after the first-order influence of ocean temperature on melt has been removed.

Removing the first-order effects of ocean forcing from the basal melt rate distribution requires a model of the relationship between ocean temperature and observed melt rates. Several such models have been proposed, and have generally assumed a linear to quadratic relationship between ocean temperature and ice shelf melt rates (Holland et al., 2008). However, estimates determined empirically or through numerical models vary widely, likely due to influences such as basal slope (Little et al., 2009) and basal roughness (Gwyther et al., 2015), which may not be the same for all ice shelves. Here, we use data from Getz to develop only the simplest possible relationship between ocean temperature and melt rates, then we investigate where and how melt observations deviate from the simple first-order model.

To relate the observed melt rates to ocean forcing, we obtain temperature profiles from 25 CTD casts taken within 6 km of Getz. We converted in situ temperatures to pressure- and salinity-dependent temperatures above freezing using the Gibbs-SeaWater Oceanographic Toolbox (McDougall and Barker, 2011). The 25 profiles of $\text{T}-\text{T}_{\text{freeze}}$ are shown in Fig. 2c. The mean profile of $\text{T}-\text{T}_{\text{freeze}}$ was then used to interpolate the local temperature above freezing corresponding to the depths of the basal ice in each grid cell of Getz. Ice basal depths were calculated assuming hydrostatic equilibrium for ice of 917 kg·m$^{-3}$ density in seawater of 1028 kg·m$^{-3}$ density, using REMA surface elevations (Howat et al., 2019) that we converted to the GL04C geoid (Förste et al., 2008; Greene et al., 2017) and from which we removed modeled firn air content (Ligtenberg et al., 2011). The resulting estimated basal temperature distribution is shown in Fig. S6.

# 3 Results

## 3.1 The new Getz bathymetry

The new map of airborne gravity-derived bathymetry is shown in Fig. 2a. The inversion reveals deep troughs are continuous from the inner continental shelf to beneath the ice shelf. In Western Getz we identify a 1300 m deep trough between Siple Island and Dean Island, which we refer to as Siple-Dean Trough (SDT). In Eastern Getz we find a 1200 m deep trough between Duncan Peninsula and Wright Island, which we refer to as Duncan-Wright Trough (DWT). Published shiptrack bathymetry (Nitsche et al., 2007) shows that the Dotson-Getz Trough on the inner continental shelf extends to the ice front, and our results suggest that DWT is the continuation of the Dotson-Getz Trough, providing a pathway for CDW to enter to the ice shelf cavity without obstruction. Similarly, SDT is the continuation of the the West Getz Trough in which unmodified CDW has been reported (Assmann et al., 2019). We note, however, that despite the similar depths and close proximity of DWT to SDT, the two troughs are not connected, but are separated by a bathymetric sill that rises to a depth of approximately $500 \pm 240$ m between Siple and Carney islands (Fig. 2a) (see methods for uncertainty estimation of the gravity inversion). The free-air gravity field also reflects the general shape of the bathymetric features. Along the profile XYZ (Fig. 2c), the bathymetric sill has higher gravity anomaly values. The trough area have low gravity anomaly values.

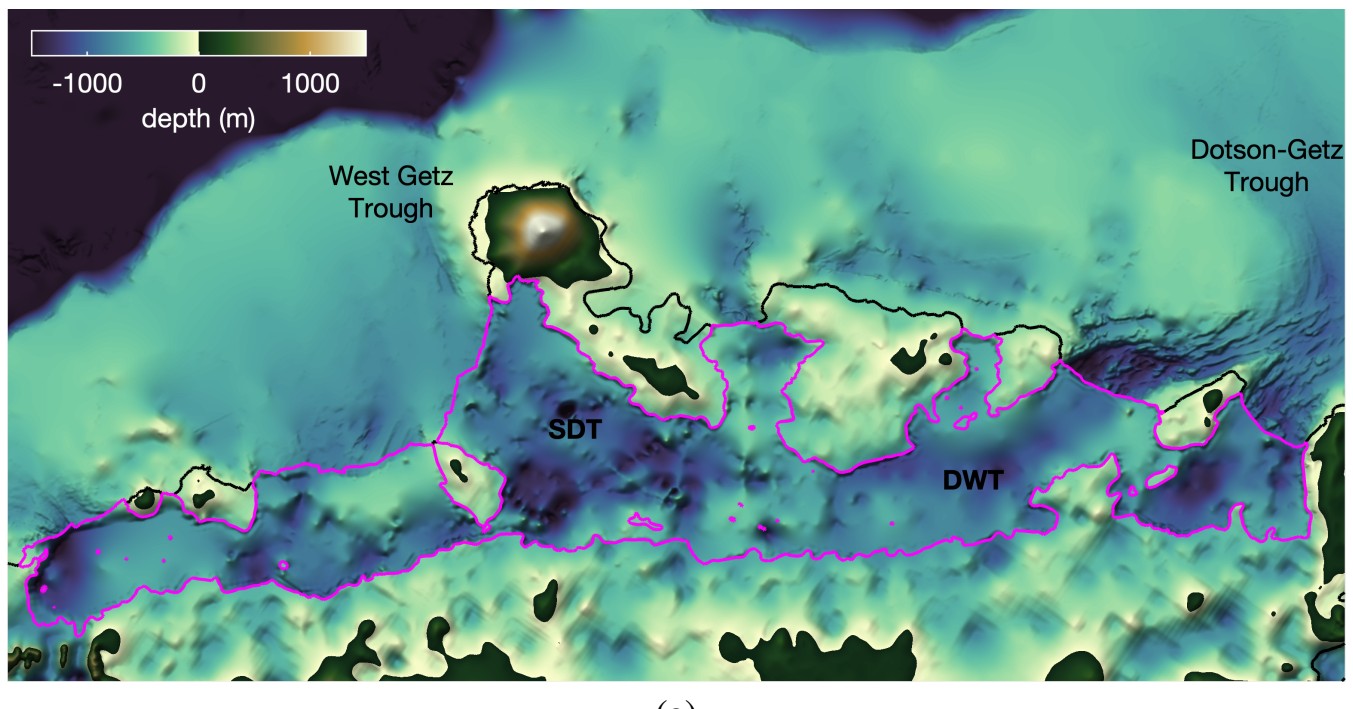

(a)

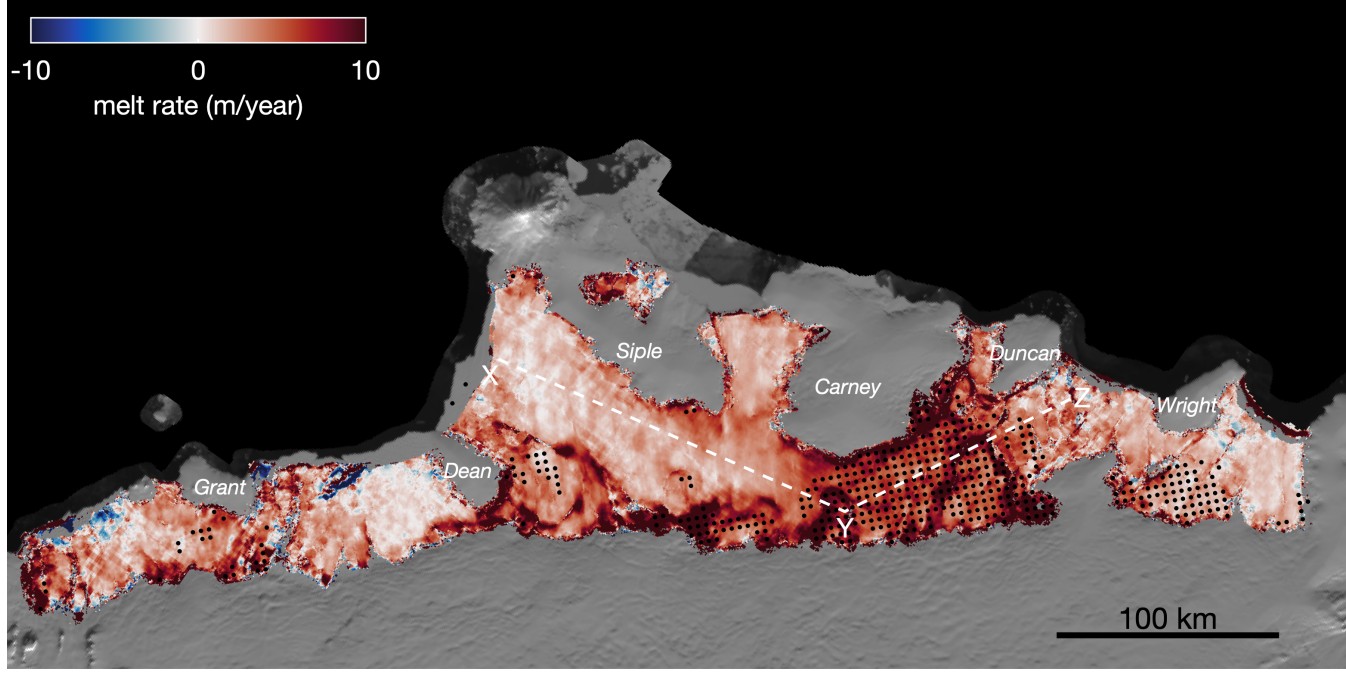

(b)

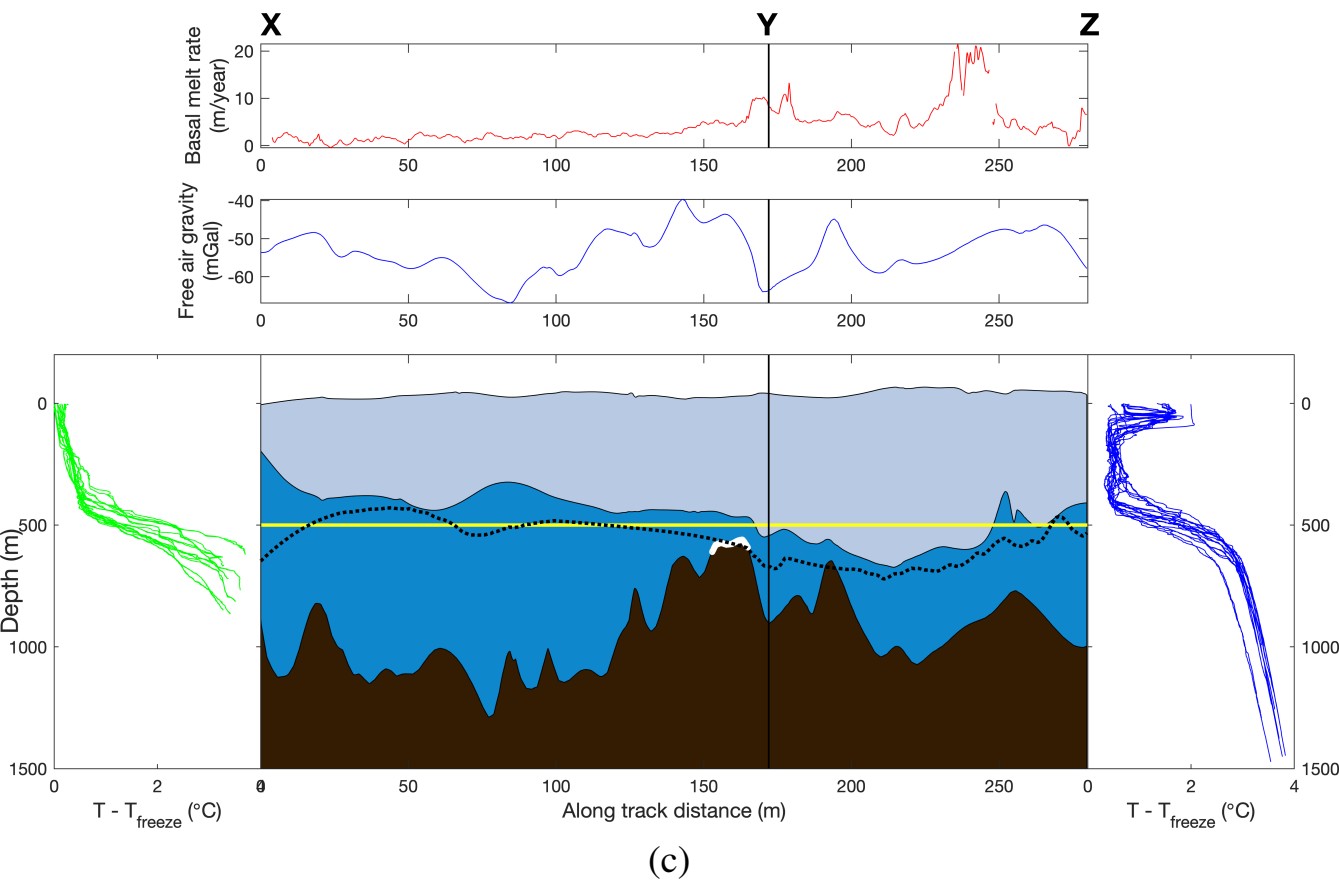

(c)

**Figure 2. The shape of sea floor, basal melt rates and along profile view of the study area.** (a) The bathymetry of the Getz Ice Shelf. The profile XYZ crosses the trough between Dean Island and Siple Island, the bathymetric sill, and the the trough between Duncan and Wright Islands. The purple is the ice shelf outline (Mouginot et al., 2017). The bathymetry of the continental shelf is from IBCSO (Arndt et al., 2013). (b) The observed basal melt rate. Black stippling indicates ice bottom elevations deeper than 500 m. The background is the MODIS-derived Mosaic of Antarctica (MOA) (Scambos et al., 2007). (c) Upper panel shows the basal melt rate along profile XYZ. The lower panel shows the elevations of ice and bedrock along profile XYZ, with the depth temperature profiles from Western and Eastern Getz. Ocean is blue, ice is light blue, and the bedrock is brown. The white indicates the location of the bathymetric sill. The black dash line is the bathymetry from Bedmap2 (Fretwell et al., 2013). The horizontal yellow line indicates the mean thermocline depth. The thermal forcing $T - T_{freeze}$ is calculated from CTD casts (Locarnini et al., 2013).

## 3.2 The melt rates of Getz

### 3.2.1 Melt rate from observation

Fig. 2b shows our observation of mean basal melt rates from 2010 to 2016. We discover that melt is concentrated along the grounding line especially where it intersects deep troughs. The area-averaged melt under Getz is 4.15 m·yr$^{-1}$, equating to

141.17 Gt·yr$^{-1}$ of freshwater flux into the Southern Ocean. We find a continuous channelized melt pattern (shown as the dark red in Fig. 2b), from the grounding zone to Eastern Getz calving front. The profile XYZ shown in Fig. 2c is sampled along SDT, the sill, and DWT. The top of the sill sits slightly below the 500 m thermocline depth, and may therefore allow exchange of warm deep waters between the Eastern and Western Getz. Fig. 2b shows that the 500 m ice bottom elevation, represented by the stippled, marks a boundary between low and high melt rates, likely resulting from the warm waters that reside below that depth.

### 3.2.2    Melt rate with no sub-glacial discharge

To understand how subglacial discharge might affect the melt rate of Getz, we compared the spatial distribution of basal melt observations to the patterns of melt that are expected to result from a simple depth-dependent model of melt rates (Holland et al., 2008). The simple model assigns melt rates based on the ice shelf draft and a corresponding depth-dependent water temperature (see Supplementary Fig. S6), taken as the mean profile of several nearby oceanographic temperature measurements (Locarnini et al., 2013) (see methods on the non-discharge melt rate). We refer to this modeled melt distribution as the "non-discharge case" because it assumes melt is driven only by the in situ farfield ocean temperature, and does not consider any potential role of local subglacial discharge. Fig. 3 shows the difference between the non-discharge case melt rate and the observed melt rate. The areas where the observed melt rate exceeds the non-discharge melt rate (red area in Fig. 3) correspond to locations of subglacial discharge predicted by GlaDS.

### 3.3    The subglacial discharge locations v.s. the melt rate difference

GlaDS predicts subglacial discharge from several major subglacial channels that line up closely with the high melt regions at the ice shelf grounding line. Channel A near Grant Island has the largest channelized relative discharge rate of about 5.3 m$^3$·s$^{-1}$, while the channel outlets near the grounding line between Carney Island and Duncan Peninsula has relative discharge rates ranging from 1.76 to 2.4 m$^3$·s$^{-1}$. Channel B near the east of the bathymetric sill has a relative discharge rate of 1.76 m$^3$·s$^{-1}$. These channel outlets and relative discharges match up with ice shelf melt rate that are more than 10 m·yr$^{-1}$. Our work confirms previous findings (Alley et al., 2016; Le Brocq et al., 2013) showing subglacial discharge outlet locations line up well with surface channels visible in the MODIS-based Mosaic of Antarctica (MOA) image (Scambos et al., 2007)(see Supplementary Fig. S7).

## 4    Discussion

### 4.1    The continuity of the troughs

The deep troughs we find extending from the inner continental shelf to Getz and are deep enough to allow the CDW observed along the ice shelf calving front to enter the ice shelf cavity (Fig. 2c). The continuity of the troughs between the Getz cavity and the continental shelf suggests that the glaciers feeding Getz may have flowed down the deep troughs and onto the continental

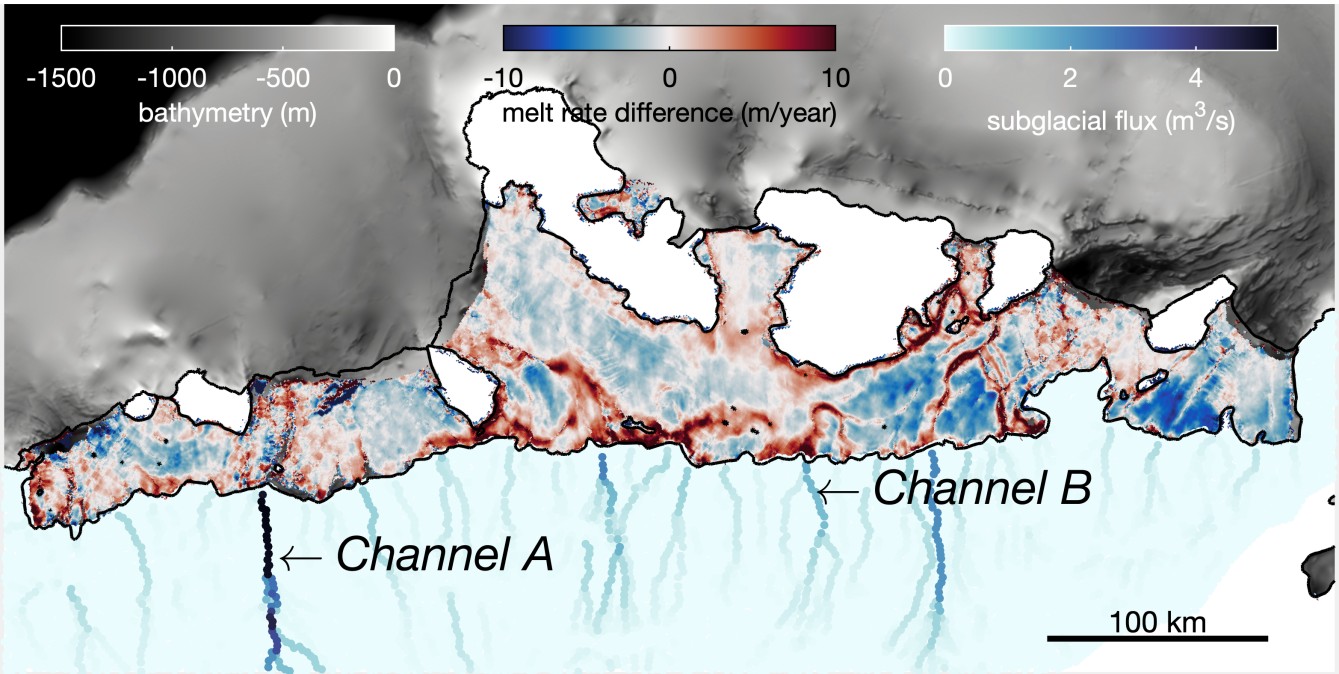

**Figure 3. The melt rate difference between observed melt rate and melt rate with no discharge.** Red indicates regions where observed melt rates are higher than can be explained by temperature-and-depth-dependent forcing alone. The blue fluxes indicates subglacial meltwater. The gray background is the bathymetry from IBCSO (Arndt et al., 2013).

shelf during the past ice age (Larter et al., 2009; Nitsche et al., 2007). The major troughs we report are not present in the publicly available Bedmap2 (Fretwell et al., 2013) or IBCSO (International Bathymetric Chart of the Southern Ocean) (Arndt et al., 210 2013) and the bathymetric sill we observe is not represented in RTOPO2 (Schaffer and Timmermann, 2016). One exception is the deep trough that is identified east of Wright Island, where the depth of trough is 200 m shallower than the trough of the inner continental shelf near the ice shelf front. This new bathymetry will provide important boundary conditions for numerical ocean modeling efforts designed to improve our understanding of ocean heat delivery to coastal ice shelves.

## 4.2 Impact of the ice draft on the melt rate

Previous oceanographic surveys have shown that the Getz melt rate is sensitive to ocean temperature, thermocline depth, circulation strength, bathymetry, and ice thickness (Jacobs et al., 2013). In our study, the factors that may affect the melt rate over the trough area are bathymetry, ice bottom elevation, incursion of warm water, subglacial discharge drained across the grounding line, and the continuity of the troughs from the grounding line to the ice shelf edge (Fig. 2a and 2b). Differences in melt regimes are apparent between the two troughs we report. Most notably, ice in the DWT experiences a much higher 220 melt rate than ice in the SDT, likely because the deep draft of the Eastern Getz places it in warm CDW, whereas the shallow base of the ice to the west sits in relatively cooler water. In Eastern Getz, the high basal melt region over DWT corresponds

to thick ice, where the base sits in the water below the 500 m thermocline depth (stippled region in Fig. 2b). In addition, the deep trough that is identified east of Wright Island does not correspond to a high melt rate. There is no pathway for CDW intrusion to the ice shelf cavity over that deep trough since the trough is not continuous from the inner continental shelf to the ice shelf cavity. Therefore, the deep trough that lies east of Wright Island is not associated with a major basal melting although the corresponding ice draft is deep (Fig. 2b).

The existence of a trough does not necessarily indicate that the melt rate will be high over it. XY is overlain by a deep trough and allows the incursion of CDW with high melt rate at the grounding line. However, the ice draft is shallower than the thermocline depth, so we do not observe high melt rate all along the trough overlain by XY. Similarly, the melt rate is high over DWT since the ice draft is deeper than the thermocline depth. In addition, subglacial hydrological modeling (Figure 3) suggests that the subglacial meltwater from upstream may drain through channel B and enhance the melt rate near Y. Therefore, although Y has a relatively shallow bathymetry, we observe a melt rate peak around Y.

### 4.3 Impact of the subglacial discharge on the melt rate

The map of basal melt rate shows several areas of localized high values along channel-like structures connected to the grounding lines. Analysis of subglacial discharge shows a striking connection between predicted channel outlets and high basal melt rates, suggesting that subglacial discharge plays a significant role in regulating the basal melt rate in Getz. Several of the channel outlet locations predicted by GlaDS correspond to ice shelf melt rates that are more than $10 \ \mathrm{m \cdot yr^{-1}}$ higher than can be explained by thermal ocean forcing alone (Fig. 3).

Subglacial discharge has been shown to increase basal melt by initiating convective cells carrying heat from warm ocean water below the thermocline to the underside of ice shelves and calving fronts (Jenkins, 2011; Slater et al., 2015). This correspondence is because the subglacial meltwater from upstream drains across the grounding line and induces large but localized sub-ice-shelf melt rates beneath the ice shelf (Le Brocq et al., 2013). One notable exception is Channel A, which pumps more subglacial discharge into the cavity than any other source, yet ice shelf melt rates here are not anomalously high. This is likely due to the presence of a bathymetric high (Fig. 2a) that prevents CDW from entering the Getz cavity to the west of Dean Island. As a result, buoyant subglacial discharge from Channel A does not entrain warm water into its plume or cause elevated channelized melt rates west of Dean Island.

### 5 Conclusions

Our new bathymetry of the Getz Ice Shelf reveals troughs that are continuous from the inner continental shelf to the ice sheet grounding line, which provide natural pathways for CDW to enter into the ice cavity and drive rapid basal melt. We show discharge of subglacial freshwater plays a significant role in regulating the basal melt rate of Getz. Our results confirm the importance of bathymetry and subglacial discharge for understanding ocean forcing on basal mass loss of Antarctic ice shelves. Our study demonstrates the practical use of high-resolution ship-borne helicopter gravity to fill critical gaps in seafloor bathymetry in Antarctica, especially over the deep troughs under the ice-shelf cavity that generally go undetected in more

regional aerogeophysical surveys. These new data will be critical for guiding new airborne/ground-based surveys, interpreting recent and past ice-shelf changes, and informing ocean circulation modeling of future impacts for this sector of West Antarctica. The controls from bathymetry and subglacial discharge on the ice shelf basal melting we have found here is likely widespread around Antarctica. Therefore, a similar study over other massive ice shelves similar to Getz should be addressed in the future.

*Data availability.* The IceBridge gravity and radar data were obtained from https://nsidc.org/icebridge/portal/. Helicopter gravity data will be deposited at https://gcmd.nasa.gov/. The CTD casts were obtained from https://www.nodc.noaa.gov/OC5/woa13/. The CryoSat-2 satellite altimetry data are available at https://earth.esa.int/web/guest/data-access. The ice velocity data were obtained from https://nsidc.org/data/nsidc-0484/. The derived data products in this paper is posted at https://doi.org/10.5281/zenodo.2527237.

*Author contributions.* W.W. performed the gravity inversion and wrote the manuscript; J.S.G assisted with mapping and gravity inversion processing; N.G performed the observed melt rate calculation; C.F.G. conducted the hydrological model study; C.A.G. estimated non-discharge case melt rate and assisted with Antarctic Mapping Tool; D.D.B. and D.A.Y. supported with the geophysical interpretations; A.W., and K.A. contributed the oceanographic inputs; T.G.R., S.H.L., T.W.K. and W.S.L. contributed to the helicopter gravity data collection. All authors contributed comments to the interpretation of results and preparation of the final paper.

*Competing interests.* The authors declare that they have no competing financial interests.

*Acknowledgements.* This work was supported by the NSF project (grant PLR-1543452), G. Vetlesen Foundation, and UTIG Gale White Fellowship. The non-discharge case melt rate research was carried out at the Jet Propulsion Laboratory, California Institute of Technology, under a contract with the National Aeronautics and Space Administration. Work on the observed melt rate was funded by European Space Agency's Support to Science Element programme through CryoTop Evolution project 4000116874/16/I-NB (NG). We thank KOPRI and Helicopter New Zealand for collecting the helicopter gravity data across the Amundsen Sea Sector. Helicopter gravity survey is supported by the Korean Ministry of Oceans and Fisheries (KIMST20190361; PM19020) and KOPRI PE18060. We also thank Geosoft Education Program for sponsoring the software to conduct the bathymetry inversion. This is UTIG contribution ####.

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
