# Peer review of "Getz Ice Shelf melt enhanced by freshwater discharge from beneath the West Antarctic Ice Sheet"

_The Cryosphere, 2019_

## Referee Comment (RC1) · Anonymous Referee #1 · 18 Aug 2019

The authors showed the results of bathymetry under Getz with ice shelf draft. They also confirm the importance of subglacial discharge for controlling basal melt rate. Authors present exciting findings with a few minor corrections needed before publication.

Minor comment

Why ice shelf draft is not shown? I would like to see ice shelf draft plotted in Figure 2 as well. I believe the additional discussion on the difference between previous and new dataset would be great.

Other comments Line 110: Is this anomaly? Authors should phrase this better as in the caption of Figure 3.

Line 127-132: Authors mention that they use the least square method to infer 3.8m /yr

[Figure]

/K. What is this value? It would be good if authors can clarify which value they try to calculate. It would be good to add an equation in the text and specify.

The caption of Figure 2: I believe you do not mark only 500 m. What is the range?

Line 154 and some other lines: Authors can not say this is Meltrate with no discharge. This is confusing. What authors did is to remove the effect of ocean temperature.

Acknowledgment: I believe authors should mention where these datasets can be downloaded. Raw data and bathymetry data.

---

## Referee Comment (RC2) · Anonymous Referee #2 · 1 Oct 2019

This paper sets out to define the sub-ice-shelf bathymetry beneath the Getz Ice Shelf using a combination of Operation ICE Bridge (OIB) and new helicopter-borne gravity data. It goes on to calculate ice-shelf basal melt rates and discuss the links to subglacial bathymetry and freshwater discharge from beneath the grounded ice sheet.

The overall concept of this paper is good, and the correlation of predicted onshoremelt pathways and ice shelf channels is striking. However, recovery of sub-ice-shelf bathymetry is challenging. A number of points regarding the inversion method should therefore be addressed and clarified before publication. In addition the assertion that "the pattern of basal melt is correlated with bathymetric troughs" and that ice-shelf basal melt is concentrated in "deep troughs" is not clearly backed up by the presented figures.

[Figure]

Specific comments: L61-62 "The gravity anomaly follows the topography in this region quite well (Fig. S2), which suggests the effectiveness of combining OIB and helicopter data".

Fig. S2 shows the gravity data overlain by the satellite imagery. It therefore does not strictly show that the gravity anomaly follows the topography. Rather it shows that the gravity highs are clearly associated with regions of ice shelf grounding and more elevated islands within the ice shelf system.

In addition this image (Fig. S2) does not really show the effectiveness of combining OIB and helicopter data. A shaded relief map of the free air gravity data with no satellite overlay would be significantly more revealing of the quality of the join between the OIB and helicopter data. I would strongly recommend adding this at least to the supplementary material. The subsequent cross-over analysis does confirm the two data sets match well, but this cannot be seen from the presented figure.

Section 2.2 Bathymetric Inversion Approach.

Given that 2D modelling underpins the inversion approach this section should be expanded. Firstly at least some of the constructed 2D models should be shown as supplementary figures. These should include input gravity, constraining radar derived topography or swath derived bathymetry if available, and the resulting modelled gravity field. This would give significant confidence in the robustness of the recovered bathymetry. In addition it is not clear if all, or only some of the flight lines shown in Figure 1 were modelled, or if the models were constructed separately using the gridded gravity data. Locations of actual models could simply be highlighted in Figure 1.

It is not clear if/how the different bathymetric 2D profile models are tied together. Were the models forced to the same level at intersecting points, or were the different derived bathymetric profiles simply merged through a gridding process? This latter strategy could introduce gridding artefacts – see next point. Adding the offsets at profile intersections could be a good additional estimate of the robustness of the2D modelling

method.

How was the bathymetric grid presented created, i.e. what interpolation method was used (e.g. spline, kriging, nearest neighbour). What was the grid mesh size of the interpolated grid? Were any steps implemented to prevent aliasing and high frequency signals on the lines generating spurious patterns in the derived bathymetric grid?

Related to the point above was onshore line radar data, or offshore swath bathymetric data included in generating the final bathymetric grid (beyond constraining the 2D gravity models)? This would be my recommendation, as it will help generate a seamless transition between the 'observed' bathymetry and the bathymetric estimates derived from gravity data.

Was any pre-processing done on the gravity data to account for long wavelength geological factors such as sedimentary basins or changes in crustal thickness? Such factors might be expected in a region like the Getz Ice Shelf which is relatively close to the continent ocean boundary.

Figure 2c. In addition to the basal melt rate and bathymetry it would be good to show the free air gravity field along this profile. This would give additional confidence in the inversion result, as individual bathymetric features could be linked to observed gravity anomalies.

L75-76 "The bathymetry model is updated iteratively until the difference between modeled gravity and observed gravity values is minimized". What is the limit on this convergence? Typically this should be (at most) the error of the observed data. Allowing the model to fit the data more precisely runs the risk of over-fitting the anomalies with more extreme topographic undulations not truly supported by the data.

L79 and Fig. S3 denotes the polygon densities applied in this region. How were these densities chosen? The densities presented are significantly higher than the standard Bouguer correction (2670 kgm-3) which is generally accepted as the typical density

of upper continental crust. Use of an unreasonably high rock density could lead to underestimation of the true amplitude of the bathymetry. The presence of apparent volcanic cones on the islands would suggest a higher than usual density (depending on lithology), however, granites seen in outcrop just the west of the survey region would indicate a density closer to the continental crustal average would be suitable. Some discussion of where these numbers come from is therefore needed.

L129 Talking about the least squares relationship between ice shelf thickness and melt rate. What is the quality of this fit (e.g. r2 value)? It would be good to show the trend line over the data point cloud to allow a more intuitive assessment of this fit.

L147-148 states "We discover that melt is concentrated along the grounding line of Getz and in deep troughs". Also L185-186. Figure 2b does show melt concentrated near the grounding line. However, it is not totally clear that melt is associated with all the deep troughs identified in the gravity data. For example in Fig. 2b along profile X-Y melt rate is relatively low, but this region is underlain by a major trough. In contrast Fig. 2c shows that the peak in melt rate around Y is associated with relatively shallow bathymetry. In addition the very deep trough identified east of Wright Island is not associated with very major basal melting, despite the ice shelf being >500m deep. It is probably fine to say deep troughs are present allowing warm water to access the grounding line region, but this is not a clear correlation between melting and deep troughs.

Technical corrections:

L31 "feasibility of gravity observations from a ship at sea" would be better as "feasibility of gravity observations from a helicopter operating from a ship at sea"

L36 "By pairing location……." might be better as "By comparing locations….."

L42-43 This sentence appears to repeat itself.

L51 "non-discharge melt rates". This is the 1st introduction of this term, which I did

not find self-explanatory. I would suggest either defining it here, or simply saying: "The observed basal melt rates were computed using a mass conservation approach from Jenkins (1991) and Gourmelen et al. (2017b), and corrected for melting driven by warm ocean waters using ice bottom elevation and nearby ocean temperature profiles (Holland et al., 2008)".

L97 "The observed basal melt rates" might be better as "the observed ice-shelf basal melt rates" to distinguish this calculation from anything onshore.

L107/108 I would define "non-discharge melt rate" up-front at this point, as it is some-what confusing, until it is explained.

L154 Suggest subheading should be "Melt rate with no sub-glacial discharge".

L215 "Therefore, a similar study over other massive ice shelves such as Getz should be addressed in the future" might be better as "Therefore, a similar study over other massive ice shelves similar to Getz should be addressed in the future".

All maps – it would be useful to have the coastlines of the islands shown, not just the edge of the ice shelf.

---

## Author Comment (AC1) · 14 Nov 2019

We thank the reviewers for their very helpful reviews. We hope that we have satisfactorily responded to all comments. We have complied both responses to the reviews into one document. Please find attached our responses to reviews, revised manuscript and supporting information, and a version of the manuscript with tracked changes.

Please also note the supplement to this comment:
https://www.the-cryosphere-discuss.net/tc-2019-170/tc-2019-170-AC1-supplement.zip

---

## Author Response (AR1)

We thank two anonymous reviewers for their suggestions that have led to a number of minor to critical improvements. Both reviewers noted that the scientific concept of the original manuscript is good in general and the connection between melt and subglacial discharge is striking.

We are grateful to reviewer 1 for the minor comments. Reviewer 1 felt "Authors present exciting findings with a few minor corrections needed before publication." We've addressed all the concerns from reviewer 1.

We appreciate reviewer 2's suggestions to expand Section 2.2 Bathymetric Inversion Approach. Reviewer 2 felt "Given that 2D modeling underpins the inversion approach this section should be expanded." Therefore, we have added new paragraphs and figures accordingly. The main updates are:

- More details about the bathymetry processing including how the polygon densities are chosen, how the bathymetry grid was merged, and description of the final bathymetric grid.
- New paragraphs on sediment uncertainty.
- New 2D gravity model profiles included as supplementary figures (Fig. S4).
- Free-air gravity field added to Figure 2c of the main text.
- Gravity anomaly (Fig. S1) in a shaded relief view.
- Coastlines of islands added to most of the maps.

We feel that the changes we made in response to the reviewers' comments have strengthened the paper and made the manuscript more readable. We hope that we have satisfactorily responded to all comments, which can be found here below. The original reviewer comments appear in *blue italics* and our response is provided in upright black text. Where line numbers are given, they refer to the version of the manuscript with changes marked on.

**RC1: Anonymous Referee #1:**

**General comments:**

*The authors showed the results of bathymetry under Getz with ice shelf draft. They also confirm the importance of subglacial discharge for controlling basal melt rate. Authors present exciting findings with a few minor corrections needed before publication.*

Thank you. Please find below a few minor corrections.

**Minor comment:**

*Why ice shelf draft is not shown? I would like to see ice shelf draft plotted in Figure 2 as well. I believe the additional discussion on the difference between previous and new dataset would be great.*

The ice shelf draft is shown in the original Figure 2b. The black dots in the Figure 2b indicates ice shelf draft deeper than 500 m. The discussion on the difference between previous and new dataset can be found from Line 220, which states "The major troughs we report are not present in the publicly available Bedmap2 (Fretwell et al., 2013) or IBCSO (International Bathymetric Chart of the Southern Ocean) (Arndt 185 et al., 2013) and the bathymetric sill we observe is not represented in RTOPO2 (Schaffer and Timmermann, 2016)."

**Other comments**:

*Line 110: Is this anomaly? Authors should phrase this better as in the caption of Figure 3.*

The passage in question previously stated "Fig. 3 shows melt rate **anomalies** after the first-order influence of ocean temperature on melt has been removed." Our use of the word anomalies was incorrect. We have revised the text, which now states "Fig. 3 shows melt rate **residuals** after the first-order influence of ocean temperature on melt has been removed."

*Line 127-132: Authors mention that they use the least square method to infer 3.8m /yr /K. What is this value? It would be good if authors can clarify which value they try to calculate. It would be good to add an equation in the text and specify.*

Both reviewers identified that the paragraph which was previously on lines 127-132 was problematic. Our original intent was to include this somewhat superfluous paragraph as a means of adding extra clarity about our process, but after considering the reviewers' feedback we see that the entire paragraph was confusing, unnecessary, and a distraction from the description of the analysis. Thus, we have removed the problematic paragraph in this revised version of the manuscript.

By removing this paragraph, we no longer mention the confusing 3.8 m/yr/K dṁ/dT slope that was used to detrend the melt rate relative to basal temperature, but we note that this value is inconsequential to our findings. The detrending step was performed only for visual context, to help unmask second-order processes that are not directly related to the first-order effects of basal temperature. An error in the value we determined would merely result in shifting the average color of melt anomalies in Figure 3, but the pattern would remain the same. Thus, we agree with the reviewers that the value of 3.8 m/yr/K was confusing and needed attention, but we feel that adding more figures or equations to describe least-squares fitting would likely be more confusing and more of a distraction to readers.

*The caption of Figure 2: I believe you do not mark only 500 m. What is the range?*

We marked ice bottom elevation deeper than 500 m. We have revised the caption of Figure 2b as "(b) The observed basal melt rate. Black stippling indicates ice bottom elevations deeper than 500 m."

*Line 154 and some other lines: Authors can not say this is Meltrate with no discharge. This is confusing. What authors did is to remove the effect of ocean temperature.*

As reviewer 2 suggested, we have added a clearer definition of "melt rate with no sub-glacial discharge" up front in the corresponding method section where has the first introduction of this term. It says "We modeled the melt rates over Getz that are expected to result from the in situ farfield ocean temperature. We refer to this modeled melt rate as non-discharge melt rate through this paper since it does not consider any potential impact from local subglacial discharge." Besides, we have changed the subheading from "melt rate with no discharge" to "melt rate with no sub-glacial discharge".

*Acknowledgment: I believe authors should mention where these datasets can be downloaded. Raw data and bathymetry data.*

The links to download the raw data, bathymetry data, and all other derived data products have been documented in the section **Data Availability**.

**RC2: Anonymous Referee #2:**

**General comments:**

*This paper sets out to define the sub-ice-shelf bathymetry beneath the Getz Ice Shelf using a combination of Operation ICE Bridge (OIB) and new helicopter-borne gravity data. It goes on to calculate ice-shelf basal melt rates and discuss the links to sub- glacial bathymetry and freshwater discharge from beneath the grounded ice sheet.*

*The overall concept of this paper is good, and the correlation of predicted onshore- melt pathways and ice shelf channels is striking. However, recovery of sub-ice-shelf bathymetry is challenging. A number of points regarding the inversion method should therefore be addressed and clarified before publication. In addition the assertion that "the pattern of basal melt is correlated with bathymetric troughs" and that ice-shelf basal melt is concentrated in "deep troughs" is not clearly backed up by the presented figures.*

Thank you. We have expanded the section on the recovery of sub-ice shelf bathymetry according to the specific comments. The list of main changes is listed on page one of this report. In addition, we have rephrased the sentences on the relationship between the melt rate and the factors that affect the melt rate of Getz, including bathymetry, ice draft, and subglacial discharge (see below the responses to the specific comment on L147 -148).

**Specific comments:**

*L61-62 "The gravity anomaly follows the topography in this region quite well (Fig. S2), which suggests the effectiveness of combining OIB and helicopter data".*

*Fig. S2 shows the gravity data overlain by the satellite imagery. It therefore does not strictly show that the gravity anomaly follows the topography. Rather it shows that the gravity highs are clearly associated with regions of ice shelf grounding and more elevated islands within the ice shelf system.*

*In addition this image (Fig. S2) does not really show the effectiveness of combining OIB and helicopter data. A shaded relief map of the free air gravity data with no satellite overlay would be significantly more revealing of the quality of the join between the OIB and helicopter data. I would strongly recommend adding this at least to the supplementary material. The subsequent cross-over analysis does confirm the two data sets match well, but this cannot be seen from the presented figure.*

We agree that Fig. S2 doesn't strictly show gravity anomaly follows topography everywhere. So, we deleted the original sentence "The gravity anomaly follows the topography in this region quite well (Fig. S2)".

We've revised Fig. S2 to be a shaded relief map of free-air gravity data with no underlying satellite imagery. The shaded relief view shows the quality of combining the OIB and helicopter datasets is good over the trough area between Dean Island and Siple Island.

*Section 2.2 Bathymetric Inversion Approach.*

*Given that 2D modelling underpins the inversion approach this section should be expanded. Firstly at least some of the constructed 2D models should be shown as supplementary figures. These should include input gravity, constraining radar derived topography or swath derived bathymetry if available, and the resulting modelled gravity field. This would give significant confidence in the robustness of the recovered bathymetry. In addition it is not clear if all, or only some of the flight lines shown in Figure 1 were modelled, or if the models were constructed separately using the gridded gravity data. Locations of actual models could simply be highlighted in Figure 1.*

As the reviewer suggested, we added 2D models as supplementary figures (Fig. S4). We show gravity lines from both OIB and helicopter gravity surveys. In the upper panel, we present input gravity and resulting modeled gravity. In the lower panel, we show radar-derived ice topography and inverted bathymetry. See Fig. S4 of the supplementary information.

In addition, we highlighted the locations of actual gravity lines as black dashed lines in Figure 1.

*It is not clear if/how the different bathymetric 2D profile models are tied together. Were the models forced to the same level at intersecting points, or were the different derived bathymetric profiles simply merged through a gridding process? This latter strategy could introduce gridding artefacts – see next point. Adding the offsets at profile intersections could be a good additional estimate of the robustness of the 2D modelling method.*

Agreed. It was not clear enough how different bathymetric 2D profiles modes are tied together. The derived bathymetric profiles from each 2D model were merged through a gridding process by the minimum curvature gridding method. However,

gridding could introduce artifacts at the intersection points. We calculate the offset the profile intersection and the average offset is about 20 m. We added one paragraph on how the bathymetric grid was created - see next response.

*How was the bathymetric grid presented created, i.e. what interpolation method was used (e.g. spline, kriging, nearest neighbour). What was the grid mesh size of the interpolated grid? Were any steps implemented to prevent aliasing and high frequency signals on the lines generating spurious patterns in the derived bathymetric grid?*

As mentioned in the previous response, minimum curvature was used as the interpolation method to create the bathymetric grid. The grid mesh size of the interpolated grid is 1792 m. To de-alias, we did two things: increase the low-pass de-sampling factor and set the blanking distance to not include the over-shoot areas of the grid. We have added one paragraph on how the grid was created, stating that,

"The different 2D bathymetric profiles are merged through the minimum curvature gridding method, provided by the *Grid and Images* module from Geosoft Oasis montaj. The details of the minimum curvature method can be found in Briggs (1974). The mesh size of the interpolated grid is 1792 m. To prevent aliasing and high frequency signals, we increase the low-pass de-sampling factor (i.e., the number of grid cells that are averaged). This factor (set to 3) removes high frequency signals since it acts as a low-pass filter by averaging all point into the nearest cell. The distance between grid cells and a valid point greater than the blanking distance (set to 2000 m) are blanked out in the final grid. However, gridding could still introduce artifacts at the intersection points. We calculate the offsets at the profile intersections, and the average offset is about 20 m."

*Related to the point above was onshore line radar data, or offshore swath bathymetric data included in generating the final bathymetric grid (beyond constraining the 2D gravity models)? This would be my recommendation, as it will help generate a seamless transition between the 'observed' bathymetry and the bathymetric estimates derived from gravity data.*

Yes, IBCSO was included to help generate the final bathymetric grid. We have added the description of the final bathymetry grid, stating that "The final derived bathymetry (Figure 1a) includes the Getz Ice Shelf bathymetry from gravity inversion and offshore bathymetry from IBCSO (International Bathymetric Chart of the Southern Ocean) (Arndt et al., 2013)."

*Was any pre-processing done on the gravity data to account for long wavelength geological factors such as sedimentary basins or changes in crustal thickness? Such factors might be expected in a region like the Getz Ice Shelf which is relatively close to the continent ocean boundary.*

There was no pre-processing done on the gravity data to account for long wavelength geological factors such as a major sedimentary basin or crustal thickness gradient. Although our methods do account for local geological heterogeneity (see the response to the densities question below), our confidence that neither longer wavelength effects or shorter wavelength draping of marine sediments are significant are described in the following added paragraph.

"We do not include significant geological or sedimentary signatures in our model since we have insufficient magnetic analysis over Getz. But our methods do account for local geological heterogeneity (see Supplementary Fig. S3). Published interpretations within ASE (Amundsen Sea Embayment) imply that we should not expect a significant crustal thickness gradient or sedimentary basin lies beneath the Getz Ice Shelf (Gohl et al., 2013). Therefore, we expect sediments near the grounding line being scoured away as seen in other ice shelves of ASE (Gohl et al., 2013; Cochran et al., 2014). However, if the sediment is present, it will cause the gravity-derived bathymetry to be deeper than the actual seafloor. Our uncertainty estimates (see Supplementary Fig. S4) indicate that our gravity derived bathymetry is shallower than both the measured off-shore bathymetry and the measured bed elevation beneath the grounded ice adjacent to Getz. If we are correct and no significant geological structure underlies the Getz then the existence of sediments will shift the bathymetry to be deeper but will not change the shape of the bathymetry, and thus will not affect our conclusions."

*Figure 2c. In addition to the basal melt rate and bathymetry it would be good to show the free air gravity field along this profile. This would give additional confidence in the inversion result, as individual bathymetric features could be linked to observed gravity anomalies.*

We added free air gravity field along profile XYZ to Figure 2c. Free air gravity anomaly is relatively low along both XY and YZ, which are over the deep troughs. The free air gravity anomaly is higher over the sill area. We added this link to the corresponding result section, which states, "The free-air gravity field also reflects the general shape of the bathymetric features (Fig. 2c). Along the profile XYZ

(Fig. 2c), the bathymetric sill has higher gravity anomaly values. The trough area has low gravity anomaly values."

*L75-76 "The bathymetry model is updated iteratively until the difference between modeled gravity and observed gravity values is minimized". What is the limit on this convergence? Typically this should be (at most) the error of the observed data. Allowing the model to fit the data more precisely runs the risk of over-fitting the anomalies with more extreme topographic undulations not truly supported by the data.*

We have added the convergence limit to the original sentence, which states, "The bathymetry model is updated iteratively until the difference between modeled gravity and observed gravity values is minimized (convergence limit = 0.1 mGal, 0.1 mGal is the standard error of observed gravity data)."

*L79 and Fig. S3 denotes the polygon densities applied in this region. How were these densities chosen? The densities presented are significantly higher than the standard Bouguer correction (2670 kgm-3) which is generally accepted as the typical density of upper continental crust. Use of an unreasonably high rock density could lead to underestimation of the true amplitude of the bathymetry. The presence of apparent volcanic cones on the islands would suggest a higher than usual density (depending on lithology), however, granites seen in outcrop just the west of the survey region would indicate a density closer to the continental crustal average would be suitable. Some discussion of where these numbers come from is therefore needed.*

One of the challenges of inverting gravity data for bathymetry models is that the densities of any geology below the water bottom are unknown. We follow the strategy of Greenbaum (2015) by first obtaining density models of bedrock from survey lines over grounded-ice areas. We then use these densities to invert for bathymetry models for survey lines over the floating-ice areas adjacent to the grounded ice density determinations. The assumption of this approach is that no significant difference in bedrock densities exist in adjacent grounded-ice and floating-ice areas. Therefore, we have added new paragraphs on where the polygon density values come from, which states,

"We first use the gravity data from grounded ice lines to invert for bedrock densities. For those areas covered by the grounded ice lines, we assume a three-layer model: a solid ice layer with density of 0.917 $g/cm_3$ of known thickness over a bedrock layer, whose density is our free parameter; the third layer is the upper

mantle with a density of 3.3 g/cm$_3$ at a depth of 20 km. The top, bottom, and thickness of the ice layer is obtained from OIB measurement and thus fixed throughout the inversion. We start the inversion with an initial guess of granitic rock density value 2.75 g/cm$_3$ since west of the survey area has granite outcrop (Mukasa et al., 2000).

The gravity data from floating ice lines is used to invert for the bathymetry under Getz. We use a four-layer model: the first layer is an ice layer with density of 0.917 g/cm$_3$ with a known depth; the second layer is a seawater layer with a density of 1.03 g/cm$_3$, the depth of this layer is our free parameter; the third layer is a bedrock layer with density inferred from grounded-ice-line gravity data; the fourth layer is the upper mantle with a density of 3.3 g/cm$_3$ at a depth of 20 km. The lines are processed one by one starting from those that are closer to grounded ice lines."

*L129 Talking about the least squares relationship between ice shelf thickness and melt rate. What is the quality of this fit (e.g. r2 value)? It would be good to show the trend line over the data point cloud to allow a more intuitive assessment of this fit.*

Reviewer 1 also identified that this paragraph needed attention (above: Line 127-132). As we describe in the response above, we have addressed the issue by removing this problem paragraph from the text.

*L147-148 states "We discover that melt is concentrated along the grounding line of Getz and in deep troughs". Also L185-186. Figure 2b does show melt concentrated near the grounding line. However, it is not totally clear that melt is associated with all the deep troughs identified in the gravity data. For example in Fig. 2b along profile X-Y melt rate is relatively low, but this region is underlain by a major trough. In contrast Fig. 2c shows that the peak in melt rate around Y is associated with relatively shallow bathymetry. In addition the very deep trough identified east of Wright Island is not associated with very major basal melting, despite the ice shelf being >500m deep. It is probably fine to say deep troughs are present allowing warm water to access the grounding line region, but this is not a clear correlation between melting and deep troughs.*

Agreed. Only some of the troughs in the Getz area correspond to a high melt rate, but high melt rate is often observed where these troughs intersect the grounding line. We have changed the original sentence from L147-148 from "We discover that melt is concentrated along the grounding line of Getz and in deep troughs." to

"We discover that melt is concentrated along the grounding line, especially where it intersects deep troughs."

To have a clearer discussion of the relationship between melt rate and the factors that impact the melt rate, we have also rephrased the original L185-186 of section 4.2. Also, we have added an explanation of why Fig. 2c shows a melt rate around Y, which is associated with shallow bathymetry. These updates are reflected in the following paragraphs, which states,

"Previous oceanographic surveys have shown that the Getz melt rate is sensitive to ocean temperature, thermocline depth, circulation strength, bathymetry, and ice thickness (Jacobs et al., 2013). In our study, the factors that may affect the melt rate over the trough area are bathymetry, ice bottom elevation, incursion of warm water, subglacial discharge drained across the grounding line, and the continuity of the troughs from the grounding line to the ice shelf edge (Fig. 2 and Fig. 3). Differences in melt regimes are apparent between the two troughs we report. Most notably, ice in the DWT experiences a much higher melt rate than ice in the SDT, likely because the deep draft of the Eastern Getz places it in warm CDW, whereas the shallow base of the ice to the west sits in relatively cooler water. In Eastern Getz, the high basal melt region over DWT corresponds to thick ice, where the base sits in the water below the 500 m thermocline depth (stippled region in Fig. 2b).

The existence of a trough does not necessarily indicate that the melt rate will be high over it. XY is overlain by a deep trough and allows the incursion of CDW with high melt rate at the grounding line. However, the ice draft is shallower than the thermocline depth, so we do not observe high melt rate all along the trough overlain by XY. Similarly, the melt rate is high over DWT since the ice draft is deeper than the thermocline depth. In addition, subglacial hydrological modeling (Figure 3) suggests that the subglacial meltwater from upstream may drain through channel B and enhance the melt rate near Y. Therefore, although Y has a relatively shallow bathymetry, we observe a melt rate peak around Y.

Besides, we have added new explanation of why the deep trough which lies east of Wright Island does not appear a high melt rate to section 4.1 (Continuity of the troughs) and 4.2. The new sentences added to section 4.1 states,

"One exception is the deep trough that is identified east of Wright Island, where the depth of trough is ~200 m shallower than the trough of the inner continental shelf near the ice shelf front."

The new sentences added to section 4.2 states,

"In addition, the deep trough that is identified east of Wright Island does not correspond to a high melt rate. There is no pathway for CDW intrusion to the ice shelf cavity over that deep trough since the trough is not continuous from the inner continental shelf to the ice shelf cavity. Therefore, the deep trough that lies east of Wright Island is not associated with a major basal melting although the corresponding ice draft is deep (Fig. 2b)."

**Technical corrections:**

*L31 "feasibility of gravity observations from a ship at sea" would be better as "feasibility of gravity observations from a helicopter operating from a ship at sea"*

Changed as suggested.

*L36 "By pairing location. . .. . .." might be better as "By comparing locations. . ..."*

Changed as suggested.

*L42-43 This sentence appears to repeat itself.*

The repeated sentence is deleted.

*L51 "non-discharge melt rates". This is the 1st introduction of this term, which I did not find self-explanatory. I would suggest either defining it here, or simply saying: "The observed basal melt rates were computed using a mass conservation approach from Jenkins (1991) and Gourmelen et al. (2017b), and corrected for melting driven by warm ocean waters using ice bottom elevation and nearby ocean temperature profiles (Holland et al., 2008)".*

We adopted the reviewer's latter suggestion. Here we simply say "The observed basal melt rates were computed using a mass conservation approach from Jenkins (1991) and Gourmelen et al. (2017b), and corrected for melting driven by warm ocean waters using ice bottom elevation and nearby ocean temperature profiles (Holland et al., 2008)." The term "discharge melt rate" is defined up-front in its corresponding section – see below the response to comment on L107/108.

*L97 "The observed basal melt rates" might be better as "the observed ice-shelf basal melt rates" to distinguish this calculation from anything onshore.*

Changed to "the observed ice shelf basal melt rate".

*L107/108 I would define "non-discharge melt rate" up-front at this point, as it is some- what confusing, until it is explained.*

Agreed. "Non-discharge melt rate" is defined here up-front: "We modeled the melt rates over Getz that are expected to result from the in situ farfield ocean temperature. We refer to this modeled melt rate as non-discharge melt rate through this paper since it does not consider any potential impact from local subglacial discharge."

*L154 Suggest subheading should be "Melt rate with no sub-glacial discharge".*

We have changed the subheading to "Melt rate with no sub-glacial discharge".

*L215 "Therefore, a similar study over other massive ice shelves such as Getz should be addressed in the future" might be better as "Therefore, a similar study over other massive ice shelves similar to Getz should be addressed in the future".*

Changed as suggested.

*All maps – it would be useful to have the coastlines of the islands shown, not just the edge of the ice shelf.*

We have updated almost all maps with the coastlines of the islands except Fig. 2b, in which islands are identifiable from the background satellite image and are labeled with their names.

[revised manuscript text omitted]

---

## Author Response (AR2)

We thank Kenichi Matsuoka and our anonymous reviewer for their thoughtful reading and comments to the manuscript. We have addressed all of the concerns. Please see below for the mark-up manuscript.

[revised manuscript text omitted]